# The impact of parkrun on life satisfaction and its cost-effectiveness: A six-month study of parkrunners in the United Kingdom

**Steve Haake**[1]*, **Helen Quirk**[2◉], **Alice Bullas**[1◉]

**1** The Advanced Wellbeing Research Centre, Sheffield Hallam University, Sheffield, The United Kingdom,
**2** Sheffield Centre for Health and Related Research (SCHARR), School of Medicine and Population Health, The University of Sheffield, Sheffield, The United Kingdom

◉ These authors contributed equally to this work.
* s.j.haake@shu.ac.uk

**Data Availability Statement:** The dataset used in this study is available from the University's public

## Abstract

An intervention suggested by the World Health Organisation that might increase life satisfaction is parkrun, a free, weekly, timed five kilometre run or walk. The issues with such interventions are (1) whether they impact on the life satisfaction of their participants, and (2) whether they are cost-effective. A study of 548 newly registered parkrunners were asked about their life satisfaction at baseline and six months later. A change of one life satisfaction point per year per participant was defined as one WELLBY (wellbeing adjusted life year), with a value of £13,000. Three approaches were used to estimate the additionality (added value) of parkrun: (1) by comparing a participant's number of parkruns to total activity; (2) by accounting for the participant's perceived impact of parkrun across 16 measures; and (3) combining these two methods equally. After six months, weighted, seasonally adjusted life satisfaction increased from a mean of 7.489 to 7.746, a change of 0.257. Both life satisfaction improvement and additionality were greatest for the least active. Assuming only half a year of benefit, the total value of the life satisfaction change for the 2019 parkrun population of 400,167 participants was estimated as £667.4m, with the least active accounting for almost half. Comparing to the cost of running parkrun in 2019 and using the activity, impact and combined methods for additionality, benefit-cost analysis ratios were found to be 16.7, 98.5 and 59.3 to 1, respectively. These were between 2.8 to 16.7 times that of other population-level physical activity interventions. Physical health was a mediator between activity and life satisfaction; mental health was only found as a mediator when combined with physical activity. Successful features of parkrun that might guide other interventions include its framing (role, time, place and cost) and ability to forge both strong and weak social ties.

## Background

With a quarter of the world inactive and the large economic burden this represents [1], it is important to explore how inactivity can be reduced cost-effectively at the population level.

repository (SHURDA) at the following: http://doi.org/10.17032/shu-180037.

**Funding:** This study was funded by Sheffield Hallam University (SJH and AB) and by The University of Sheffield (HQ). The funders had no role in study design, data collection and analysis, decision to publish, or preparation of the manuscript.

**Competing interests:** I have read the journal's policy and the authors of this manuscript have the following competing interests: all authors are parkrunners and members of the parkrun Research Board (SH is Chair; AB and HQ are Deputy-Chairs).

Mass participation events such as 10k races, half-marathons and marathons have been suggested as one approach [2, 3], although they do not always attract the least active and have limited potential to change behaviour of the population [2, 4, 5]. This may be because life satisfaction increases may return to previous levels relatively quickly, reducing the effectiveness of mass-events as interventions [6].

In its Global Action Plan on Physical Activity, the World Health Organization recommended that countries "*implement regular mass-participation initiatives in public spaces, engaging whole communities, to provide free access to enjoyable and affordable, socially and culturally appropriate experiences of physical activity*". It gave the parkrun initiative as an example of how to do this [1]. Parkrun is a free, weekly, timed 5 km run or walk in which participants can also volunteer. It has been operating for more than 20 years and now exists in 22 countries worldwide; it has hundreds of thousands of participants each weekend [7].

A cross-sectional study of around 60,000 parkrun participants showed that those who were previously inactive were more likely to report improvements to physical health, mental health, lifestyle and a sense of personal achievement [8]. This was also true of those with mental health conditions [9]. A scoping review on parkrun concluded that parkrun participants showed sustained improvements to fitness, physical activity levels, and body mass index with a dose–response effect for participation frequency [10]. These changes were found to last at least 12 months [11]. An Irish study on middle-aged men at parkrun found that both weak and strong social connections were made after repeated participation, and that these led to improved subjective mental wellbeing [12]. The physical and mental benefits of parkrun also extend to volunteers who never run or walk but are considered as participants [13]. However, for parkrun to be useful as a population-level physical activity intervention, it is important to understand if it is a cost-effective approach.

There is increasing consensus amongst policy makers that when determining cost effectiveness, we should pay more attention to the happiness and subjective wellbeing of citizens rather than traditional measures of success such as growth in gross domestic product (GDP) [14]. The World Happiness Report, for instance, has monitored happiness across the globe using an index composed of three questions on life evaluation, positive emotions and negative emotions [15]. The report has previously advocated the use of life satisfaction as a measure to quantify the economic value of wellbeing to allow governments to justify budgets based on wellbeing [16].

Evidence suggests that low scores of life satisfaction are related to poor health, unemployment, lack of social contact and separation from partners [17]. Some have suggested that improving life satisfaction over the long-term may not be possible since it may return to a 'set-point' related to an individual's underlying traits [18–20]. A study of 3,608 German nationals over a 17-year period found that most did indeed have relatively unchanging life satisfaction but, for almost one in ten, life satisfaction changed by at least three points on a 10-point scale [21]. Not only was change over the long-term possible, variability was greater for those with the lowest life satisfaction, suggesting they might be more receptive to external influences.

Levels of physical activity have been shown to influence life satisfaction [22] and two studies in the UK and the USA found that life satisfaction correlated positively with activity [23, 24]. The link between life satisfaction and physical activity is possibly mediated through mental and physical health [17, 22] with the ultimate outcome that risk factors for many non-communicable diseases are reduced [25].

The cost-effectiveness of sport and physical activity has predominantly been calculated using detailed analysis of downstream economic benefits to give an overall social return on investment [26]. This allows comparison with traditional economic analyses using benefit-cost ratios. A systematic review showed that for interventions aimed at a general population, the benefit-cost ratio was 5.9 to 1 [26]. If interventions were targeted to specific health conditions

or populations (e.g. those with physical or mental health problems from deprived backgrounds), the ratio rose to 44 to 1 [26]. While the social return on investment approach is both detailed and thorough, it is resource intensive. An alternative is the wellbeing-adjusted life year or WELLBY, used by the UK government and the World Happiness Report [16, 27]. One WELLBY is equivalent to a one-point change in life satisfaction per person per year. Each WELLBY is estimated to be worth £13,000, with lower and upper limits of £10,000 and £16,000 [27]. The life satisfaction question is useful in two ways: (1) it gives a subjective measure of wellbeing; and (2) it enables cost-effectiveness calculations to be carried out.

In 2018, a longitudinal survey that was sent to newly registered UK parkrunners included the life satisfaction question required to calculate cost-effectiveness using the WELLBY approach [28]. This survey gives an opportunity to ask the following two questions:

1. To what extent does parkrun participation impact on the life satisfaction of participants?

2. What is the cost-effectiveness of parkrun?

If engagement with parkrun as a runner or walker does improve life satisfaction, a secondary question is: what characteristics of parkrun contribute to this improvement?

## Materials and methods

### Ethics statement

Approval for the research was given by the Research Ethics Committee of Sheffield Hallam University on 24th July 2018 (ER7034346). Participants gave written informed consent before completion, and all procedures were in accordance with the UK Data Protection Act of 2018 and the EU General Data Protection Regulation.

### The survey

An online (Qualtrics) survey, published in Quirk et al. [23], was designed to assess the health and wellbeing of parkrun registrants (the full details of which can be found in S1 Text); methods adhere to The Checklist for Reporting Results of Internet E-Surveys (CHERRIES) [29]. Respondents could answer up to 47 questions at each time point (with the majority being optional) and were able to review and change answers until submitting the completed survey.

This study uses responses to the following questions:

1. **Life satisfaction**: "Overall, how satisfied are you with your life nowadays? where 0 is 'not at all satisfied' and 10 is 'completely satisfied'." Respondents were provided with a visual-analogue scale [30].

2. **Activity level (parkrun 4-week registration question)**: This question is asked routinely at parkrun registration and was repeated in the survey. "Over the last 4 weeks, how often have you done at least 30 minutes of moderate exercise (**enough to raise your breathing rate**)?" (survey emphasis). Allowed responses were: less than once per week; about once per week; about twice per week; about three times per week; four or more times per week; rather not say/don't know.

3. **Activity level (survey 7-day question)**: To allow a scale response for activity, this question asked. "In the past week, on how many days have you done a total of 30 minutes or more of physical activity, which was enough to raise your breathing rate. This may include sport, exercise, and brisk walking or cycling for recreation or to get to and from places, but should not include housework or physical activity that may be part of your job." Possible answers were from 0 to 7 days.

4. **Mental health status:** this was measured using the Short Warwick Edinburgh Mental Well-being Scale (SWEMWBS): "Below are some statements about feelings and thoughts. Please tick the box that describes your experience of each **over the last 2 weeks**" (survey emphasis). Respondents were given 7 statements with the following possible responses: none of the time, rarely, some of the time, often, all of the time [31].

5. **General health status**: the EQ-5D visual-analogue-scale: "We would like to know how good or bad your health is TODAY. This scale is numbered from 0 to 100 [32].

6. **Perceived impact of running or walking at parkrun**: At six months, the following question was asked: "Thinking about the impact of parkrun on your health and wellbeing, to what extent has running or walking at parkrun changed:" Respondents were presented with a randomised list of 16 impacts and were allowed the following responses: 'much worse', 'worse', 'no impact', 'better' and 'much better' and were coded as -1, -0.5, 0, 0.5 and 1 respectively.

The longitudinal survey was emailed to new parkrun registrants during the first quarter of 2019 between 14th January and 11th February 2019, and again at follow up six months later in the third quarter between 16th July and 10th August 2019.

## Matching data from parkrun and comparison with the full parkrun population

To enable the survey data to be weighted to the UK parkrun population, parkrun provided summative demographic data of all new registrants in the 2019 calendar year who subsequently completed at least one parkrun. This data included the following: age (16+); gender (male or female); index of multiple deprivation ranking derived from postcode at registration (converted to quartiles where the first quartile indicates the most deprived neighbourhoods); activity level at registration (using the question above) and total number of parkruns completed. Survey respondents provided their unique parkrun ID number (from their barcode), their date of birth and name of home parkrun to allow them to be individually matched to this data and to the number of parkruns completed.

## Seasonal effect

Since the baseline survey took place in the UK in quarter one of 2019, and the follow up survey in quarter three, a seasonal effect on life satisfaction was accounted for using weighting factors created by the ONS [33]. The quarter one and quarter three weighting factors were 1.00265 and 0.99846 (see S1 Data). These had the effect of increasing values in the winter when life satisfaction was depressed and decreasing them in the summer months when life satisfaction was enhanced.

## Estimation of life satisfaction change and additionality due to parkrun

Life satisfaction change for each participant was calculated by subtracting the value at baseline from the value at 6 months. The formal definition of a WELLBY is a one-point change in life satisfaction per person per year and thus the WELLBYs for participant $i$ was calculated as half the life satisfaction using:

$$WELLBY_i = \frac{1}{2}\left(LS_{i6\ months} - LS_{ibaseline}\right)$$ (1)

where $LS_{i6months}$ and $LS_{ibaseline}$ represent values of seasonally adjusted life satisfaction values at six months and baseline. The ½ accounts for the change over half a year.

Three methods were used to estimate the additionality due to parkrun, i.e. the proportion of the life satisfaction change in Eq 1 that might be attributed to parkrun participation.

**Method 1** assumed that any additionality due to parkrun was caused solely by participation as a runner or walker and was equal to an individual's number of parkruns as a proportion of their total activity. For participant $i$ completing $N_i$ parkruns after six months and being active $A_i$ days per week at six months (using the single-item activity question), the activity additionality was:

$$Activity\ additionality = N_i/(26 \times A_i) \qquad (2)$$

The minimum value of this ratio was one parkrun in six months (or 182.5 days) giving an additionality factor of 0.0055. A maximum value of 1 was possible if the participant did one parkrun every week for 26 weeks and selected 'one day per week' for $A_i$. WELLBYs for each participant were calculated using the following:

$$WELLBY\ (activity)_i = Activity\ additionality_i \times \frac{1}{2}\left(LS_{i6\ months} - LS_{ibaseline}\right) \qquad (3)$$

**Method 2.** Survey question 6 above assessed the perceived impact of participation in parkrun as a runner or walker. The mean score of all 16 responses was used to calculate the relative impact for each individual and was assumed to be their unique impact additionality of parkrun. For example, a participant responding 'better', or 0.5, to all 16 questions would have a mean impact of 0.5 so that only half their life satisfaction change was attributed to parkrun. WELLBYs were calculated for each participant $i$ using the following:

$$WELLBY\ (impact)_i = Impact\ additionality_i \times \frac{1}{2}\left(LS_{i6\ months} - LS_{ibaseline}\right) \qquad (4)$$

**Method 3** assumed that additionality was caused by an equal combination of activity additionality and impact additionality. WELLBYs for each participant $i$ were calculated using the following:

$$WELLBY\ (combined)_i = \frac{(Activity\ additionality + impact\ additionality)_i}{2} \times \frac{1}{2}\left(LS_{i6\ months} - LS_{ibaseline}\right) \qquad (5)$$

## Total WELLBYs and benefit-cost ratio

The weighted WELLBY for each individual was calculated using Eqs 1 to 5, allowing a mean value for each method to be determined. Total WELLBYs for 2019 were calculated by multiplying the mean for each method by the number of newly registered participants in 2019 (400,167). Benefit-cost ratios were calculated using the following:

$$Benefit\ cost\ ratio = \frac{(Total\ WELLBYs - Cost\ of\ parkrun)}{Cost\ of\ parkrun} \qquad (6)$$

The 2019 cost for operating parkrun was £4,549,499 (£2,274,749 for six months) [31].

## Statistics

**Preliminary analysis.** Data was validated using Microsoft Excel for Mac (v16.82) using standard descriptors such as counts, mean, median, quartiles, minimum, maximum, skewness and kurtosis. This resulted in 548 respondents with data at baseline and six months and

approximately 75% matched to registration data. Survey questions were optional giving varying frequencies: all counts are shown in the analysis.

Propensity weighting was used to weight the longitudinal sample to the 2019 full parkrun population using age, parkruns completed, activity at registration, index of multiple deprivation and gender (in this order). Three weighting iterations were performed (available with full dataset).

**Primary analysis.** All statistical analysis was carried out using IBM SPSS Statistics for Mac (v26). Participant characteristics for the survey were compared to the 2019 parkrun population using Chi-square tests ($X^2$), as were categorical data between baseline and follow-up. Paired continuous data between baseline and follow-up were compared using the Wilcoxon Rank Test.

Effect sizes were calculated for the $X^2$ statistic using $\phi_c = \sqrt{(\chi^2/n(k-1)}$ where $\chi^2$ was the test statistic, $n$ was the number of respondents and $k-1$ was the number of rows or columns (whichever was the smaller). For the Wilcoxon Rank Tests, effect size was calculated using $r = Z/\sqrt{n}$, where $Z$ was the standardised test statistic and $n$ the number of ranked respondents [34, 35]. Effect sizes <0.1 were considered small, between ≥0.1 and <0.2 small to moderate, and ≥0.2 moderate. The strength of correlations between life satisfaction, activity and mediating variables were categorised as weak $r<0.1$, moderate $0.1 \leq r < 0.5$ and strong ≥0.5 [34].

Mediation analysis was carried out by using partial correlations (IBM SPSS Statistics for Mac, v26) and controlling for mediation variables. Physical health and mental health were tested as mediating variables between activity level and life satisfaction, in a similar fashion to previous research [36].

## Results

### Characteristics of the survey sample compared to the 2019 parkrun population

Fig 1 shows summary data for the unweighted sample compared to the full 2019 parkrun population with data and statistical information given in Table 1. These show the following:

1. The gender distribution of the sample and the 2019 population were similar ($p = 0.122$) with just under half of the sample female (44.6% vs 48.1%; Fig 1A).

2. The age distribution of the sample was skewed towards older participants (median 44 to 49 years) compared to younger participants for the 2019 population (35 to 39 years) (Fig 1B); the two were statistically different at $p<0.001$ with a small effect size.

3. The distributions of the Index of Multiple Deprivation for the sample and the 2019 population were statistically similar ($p = 0.652$) and showed that proportions in each IMD quartile increased from around 10–12% in quartile one to 36–37% in quartile four (Fig 1C).

4. The distributions of activity at registration for the sample and the 2019 population were statistically similar ($p = 0.917$) and showed that proportions increased from around 9% for those doing <1 day per week to around 30% for those doing around 3 days per week (Fig 1D).

5. More than 60% of the 2019 population did 1 or 2 parkruns, with proportions decreasing in an exponential manner as the number of parkruns increased. In comparison, only 18% of the sample did 1 or 2 parkruns with proportions decreasing in a more linear fashion as the number of parkruns increased (Fig 1E). The distributions were significantly different ($p<0.001$).

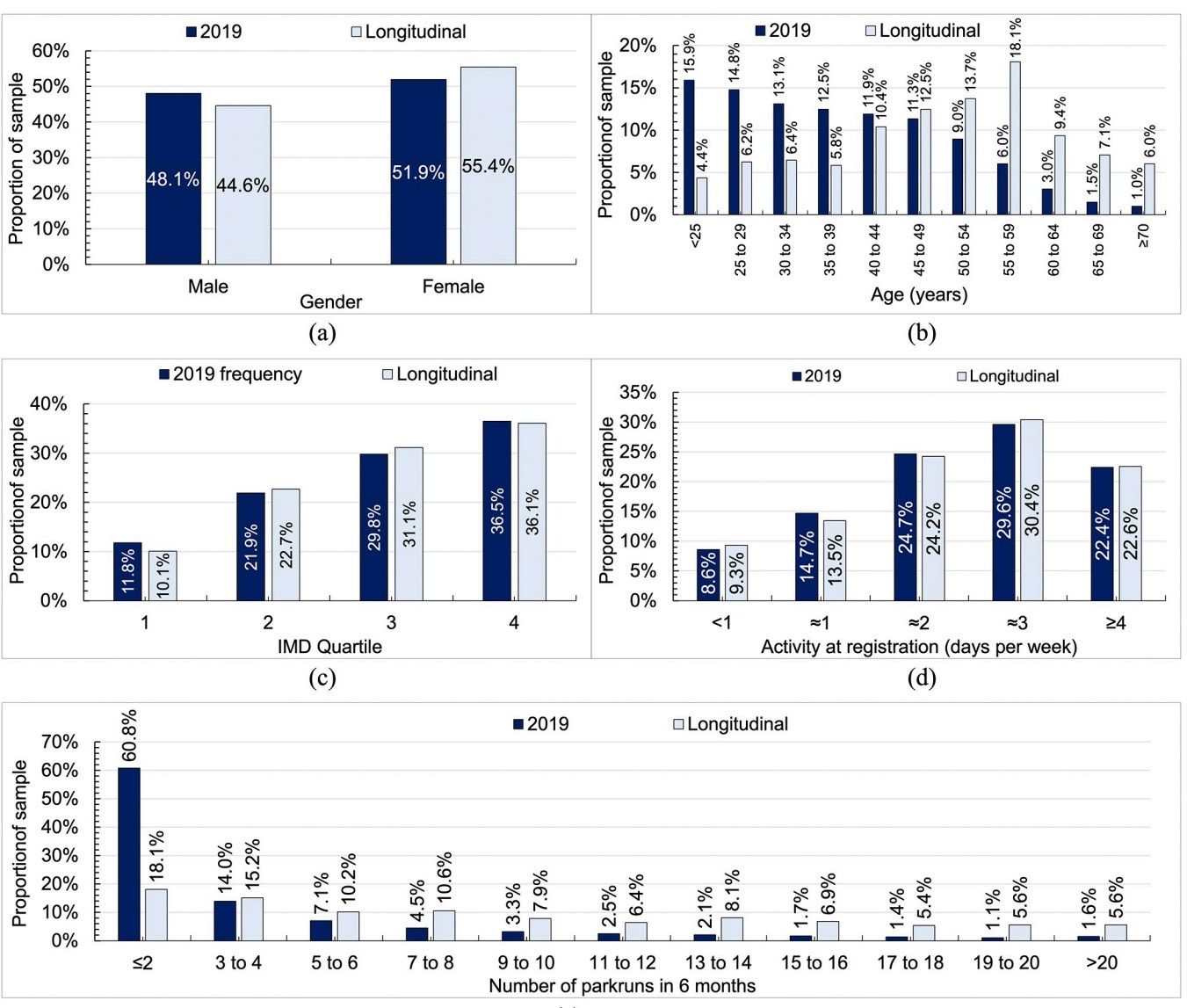

**Fig 1.** Unweighted demographics of the sample compared to the 2019 parkrun population: (a) gender; (b) age; (c) index of multiple deprivation quartile of home postcode; (d) activity at registration in days per week over the last four weeks; and (e) number of parkruns completed by 6 months.

## Changes between baseline and six months

Table 2 shows that participants had run or walked a median of eight parkruns after six months; when weighted this reduced to two parkruns. Activity level for both the weighted and unweighted samples did not change between baseline and six months with means of 3.2 to 3.4 days per week. EQ-5D VAS mean score increased significantly from baseline to six months for the unweighted sample from 78.1 to 81.6 ($p<0.001$) and from 79.0 to 81.5 for the weighted sample ($p<0.001$); the effect sizes were moderate. The SWEMWBS mean score increased between baseline and six months from 24.8 to 25.4 ($p<0.001$; small effect size) but showed no change when weighted to the full population.

The mean value of life satisfaction increased significantly between baseline and six months by 0.26 when unweighted and 0.29 when weighted ($p<0.001$); effect sizes were moderate.

**Table 1. Characteristics of the sample compared to new registrants who participated at least once in 2019.** Comparisons use the chi-square test, with effect size given by Cramer's V (Φv).

| Gender | 2019 population | | Sample | | Test statistic | p | Φv |
|---|---|---|---|---|---|---|---|
| Male | 192,379 | 48.1% | 218 | 44.6% | 2.388 | 0.122 | 0.00 |
| Female | 207,788 | 51.9% | 271 | 55.4% | | | |
| Total | 400,167 | 100.0% | 489 | 100.0% | | | |
| **Age 16+** | **2019 population** | | **Sample** | | **Test statistic** | **p** | **Φv** |
| <25 | 53,717 | 15.9% | 21 | 4.4% | 511.4 | <0.001 | 0.04 |
| 25 to 29 | 49,941 | 14.8% | 30 | 6.2% | | | |
| 30 to 34 | 44,322 | 13.1% | 31 | 6.4% | | | |
| 35 to 39 | 42,161 | 12.5% | 28 | 5.8% | | | |
| 40 to 44 | 40,188 | 11.9% | 50 | 10.4% | | | |
| 45 to 49 | 38,287 | 11.3% | 60 | 12.5% | | | |
| 50 to 54 | 30,245 | 9.0% | 66 | 13.7% | | | |
| 55 to 59 | 20,370 | 6.0% | 87 | 18.1% | | | |
| 60 to 64 | 10,304 | 3.1% | 45 | 9.4% | | | |
| 65 to 69 | 5,016 | 1.5% | 34 | 7.1% | | | |
| ≥70 | 3,350 | 1.0% | 29 | 6.0% | | | |
| Total | 337,901 | 100.0% | 481 | 100.0% | | | |
| **IMD Quartile** | **2019 population** | | **Sample** | | **Test statistic** | **p** | **Φv** |
| 1 | 42,611 | 11.8% | 49 | 10.1% | 1.634 | 0.652 | 0.00 |
| 2 | 79,022 | 21.9% | 110 | 22.7% | | | |
| 3 | 107,598 | 29.8% | 151 | 31.1% | | | |
| 4 | 131,745 | 36.5% | 175 | 36.1% | | | |
| Total | 360,976 | 100.0% | 485 | 100.0% | | | |
| **Activity level (days per week)** | **2019 population** | | **Sample** | | **Test statistic** | **p** | **Φv** |
| <1 | 33,206 | 8.6% | 45 | 9.3% | 0.950 | 0.917 | 0.00 |
| ≈1 | 56,756 | 14.7% | 65 | 13.5% | | | |
| ≈2 | 95,287 | 24.7% | 117 | 24.2% | | | |
| ≈3 | 114,336 | 29.6% | 147 | 30.4% | | | |
| ≥4 | 86,566 | 22.4% | 109 | 22.6% | | | |
| Rather not say / don't know | 6,369 | | | | | | |
| Unset | - | | | | | | |
| Total | 386,634 | 100.0% | 483 | 100.0% | | | |
| **Number of parkruns completed in 6 months** | **2019 population*** | | **Sample** | | **Test statistic** | **p** | **Φv** |
| ≤2 | 603,031 | 60.8% | 87 | 18.1% | 601.7 | <0.001 | 0.02 |
| 3 to 4 | 138,502 | 14.0% | 73 | 15.2% | | | |
| 5 to 6 | 70,267 | 7.1% | 49 | 10.2% | | | |
| 7 to 8 | 44,462 | 4.5% | 51 | 10.6% | | | |
| 9 to 10 | 32,383 | 3.3% | 38 | 7.9% | | | |
| 11 to 12 | 25,242 | 2.5% | 31 | 6.4% | | | |
| 13 to 14 | 20,858 | 2.1% | 39 | 8.1% | | | |
| 15 to 16 | 17,165 | 1.7% | 33 | 6.9% | | | |
| 17 to 18 | 14,101 | 1.4% | 26 | 5.4% | | | |
| 19 to 20 | 10,830 | 1.1% | 27 | 5.6% | | | |
| >20 | 15,421 | 1.6% | 27 | 5.6% | | | |
| Total | 992,262 | 100.0% | 481 | 100.0% | | | |

*This column represents all those who participated during 2019 regardless of when they had registered.

**Table 2. Mean, median and standard deviations of variables at baseline and six months for both weighted and unweighted sample data.** Standard deviations shown in brackets.

| Survey item | n | Unweighted Mean Median (Standard Deviation) | | | Wilcoxon Rank Test for paired comparisons | | | n | Weighted Mean Median (Standard Deviation) | | | Wilcoxon Rank Test for paired comparisons | | |
|---|---|---|---|---|---|---|---|---|---|---|---|---|---|---|
| | | Baseline | Six months | Change | Standard-ised test statistic | p | Effect size | | Baseline | Six months | Change | Standard-ised test statistic | p | Effect size |
| Number of parkruns completed at six months | 489 | | 9.05 8 (6.48) | | | | | 468 | | 4.07 2 (4.18) | | | | |
| Activity level using single item question (days per week) | 548 | 3.383 3 (1.64) | 3.318 3 (1.78) | -0.066 0 (1.76) | -1.040 | 0.298 | -0.04 | 465 | 3.245 3 (1.56) | 3.222 3 (1.82) | -0.023 0 (1.95) | -0.156 | 0.876 | -0.01 |
| EQ-5D VAS | 544 | 78.10 80 (13.13) | 81.62 85 (12.93) | 3.52 5 (11.83) | 7.437 | <0.001 | 0.32 | 463 | 79.00 80 (12.33) | 81.48 85 (12.90) | 2.478 0 (11.61) | 4.518 | <0.001 | 0.21 |
| SWEMWBS | 506 | 24.82 24.6 (4.41) | 25.44 25.0 (4.37) | 0.625 0 (3.74) | 3.775 | <0.001 | 0.17 | 429 | 24.74 25.0 (4.36) | 24.94 24.1 (4.42) | 0.208 0 (3.37) | 0.249 | 0.804 | 0.01 |
| Life satisfaction | 548 | 7.578 8 (1.55) | 7.834 8 (1.49) | 0.258 0 (1.32) | 4.702 | <0.001 | 0.20 | 465 | 7.469 8 (1.62) | 7.758 8 (1.52) | 0.288 0 (1.32) | 4.904 | <0.001 | 0.23 |
| Life satisfaction seasonally adjusted | 548 | 7.599 8.0 (1.55) | 7.822 8.0 (1.48) | 0.224 0.0 (1.32) | -0.793 | 0.428 | -0.03 | 465 | 7.489 8.0 (1.62) | 7.746 8.0 (1.52) | 0.257 0.0 (1.31) | -0.859 | 0.390 | -0.04 |
| Means by activity level <1 | 43 | 7.112 | 7.622 | 0.528 | | | | 40 | 6.626 | 7.389 | 0.768 | | | |
| ≈1 | 65 | 7.497 | 7.888 | 0.391 | | | | 69 | 7.507 | 7.931 | 0.423 | | | |
| ≈2 | 113 | 7.720 | 7.937 | 0.180 | | | | 114 | 7.422 | 7.731 | 0.297 | | | |
| ≈3 | 145 | 7.821 | 7.901 | 0.093 | | | | 130 | 7.657 | 7.709 | 0.056 | | | |
| ≥4 | 109 | 7.653 | 8.001 | 0.348 | | | | 105 | 7.656 | 7.830 | 0.174 | | | |

When life satisfaction was seasonally adjusted, the mean unweighted and weighted changes were smaller at 0.22 and 0.26 respectively and were no longer statistically significant. For context, Fig 2 compares the weighted, seasonally adjusted life satisfaction data at baseline and six months to the full UK population [32]. Life satisfaction for the sample is lower than the UK population at baseline, and higher than it after six months.

### Influence of activity level at registration

Table 2 also shows mean life satisfaction segmented by activity level at registration. The least active (<1) group had the lowest life satisfaction at baseline of 6.626 (weighted, seasonally adjusted) and the most active (≥4) the greatest at 7.656. Life satisfaction increased for each group at six months with the least active group increasing the most by 0.768 to 7.389. Fig 3A shows the life satisfaction changes in the form of box plots and suggests lower improvement in life satisfaction with increasing activity level to 0.056 and 0.174 for the two most active (≈3 and ≥4 groups).

Fig 3B shows the total weighted number of parkruns completed; this was higher for the least active (mean of 4.60 and 5.35 parkruns for <1 and ≈1 groups) and lower for the most active (mean of 3.11 parkruns for ≥4 group). Fig 3C shows total activity in days per week; 2.01 days per week for the least active (<1) group rising to 4.25 days per week for the most active (≥4) group.

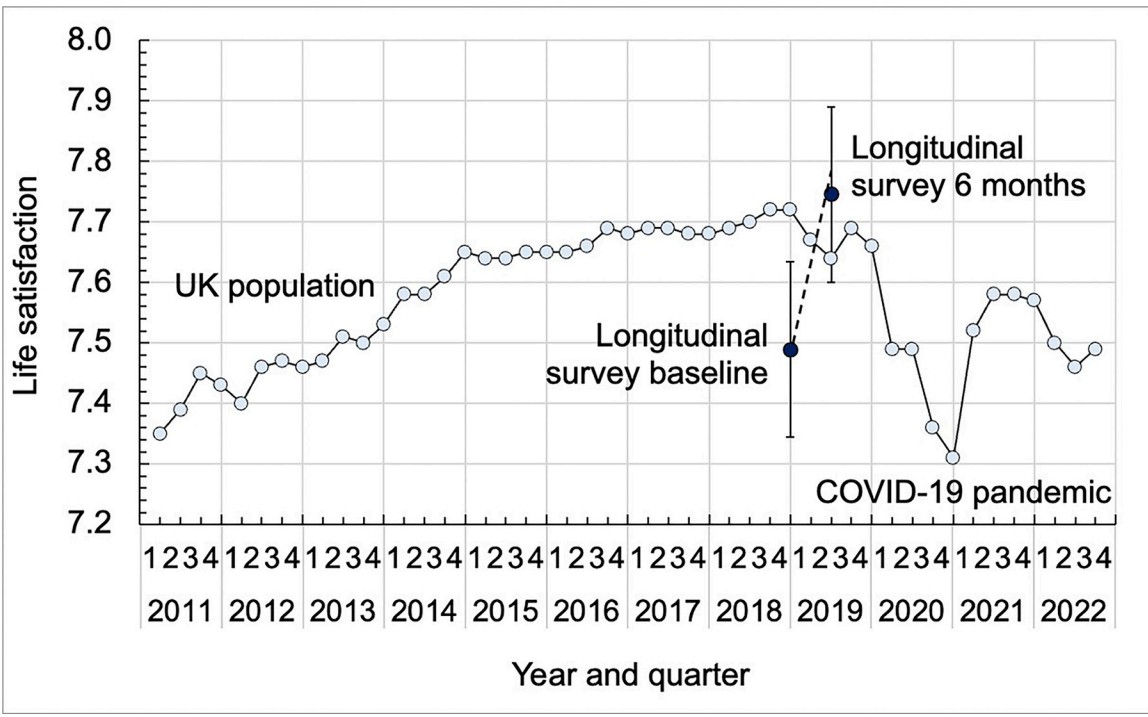

**Fig 2. Comparison of seasonally adjusted life satisfaction from the survey with the UK population [33].** Error bars show 95% confidence intervals.

### Self-reported impact of parkrun

Table 3 shows values from the 16 self-reported responses on the impact of running or walking at parkrun. Responses ranged from a mean of 0.455 for 'your sense of personal achievement' in which most participants selected at least 'better', to 0.096 for 'the amount of time you spend with family' in which most participants indicated 'no impact'. The mean for all 16 questions was 0.271, close to the median of 0.254. Health measures such as 'your fitness', 'your physical health', and 'your mental health' had values above the overall mean (0.441, 0.419 and 0.291 respectively). Measures related to social connections such as 'the amount of time you spend with family', 'the amount of time you spend with friends', 'the number of new people you meet', and 'how much you feel part of a community' had values below the overall mean (0.096, 0.149, 0.198 and 0.247 respectively).

### Additionality due to parkrun

Additionality using each of the three methods previously described are shown in Fig 4, segmented by activity level at registration. Mean weighted values for the activity, impact and combined additionality were 0.064, 0.271 and 0.172 respectively. Activity additionality in Fig 4A is highest for the least active (<1) group (mean = 0.124) and lowest for the most active (≥4) group (mean = 0.036).

Fig 4B shows impact additionality which are highest for the least active (<1) group (0.315) and lowest for the most active (≥4) group (0.250), the difference is less marked when compared to activity additionality. Fig 4C shows the combined additionality; this falls linearly from 0.225 for the least active (<1) group to 0.144 for the most active (≥4) group.

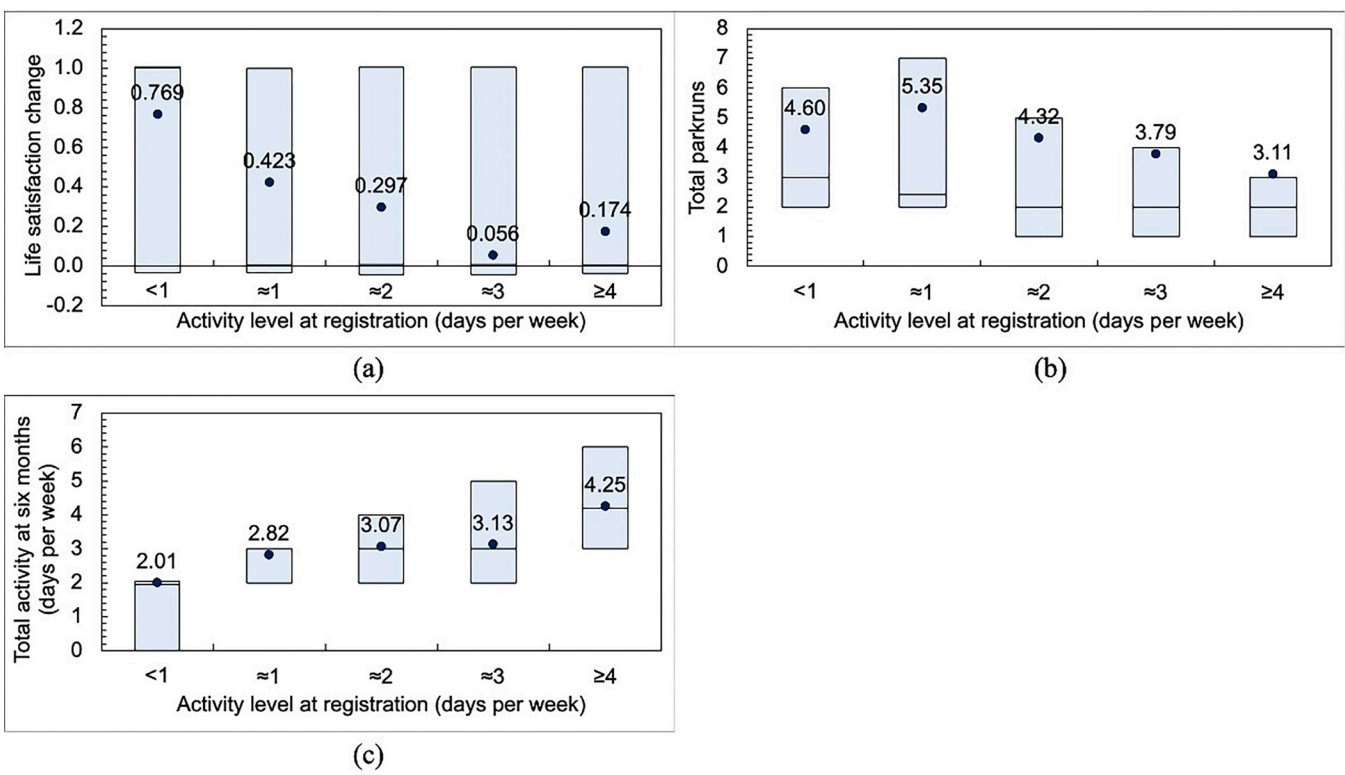

**Fig 3.** Weighted variables following six months of participation in parkrun as a runner or walker: (a) life satisfaction change; (b) total parkruns completed; and (c) total activity in days per week. Circles indicate means while boxes indicate interquartile ranges and medians.

**Table 3. Mean, standard deviation, median, minimum and maximum for impact questions relating to the impact of parkrun as a runner or walker.** The responses 'much worse', 'worse', 'no impact', 'better' and 'much better' were coded as -1, -0.5, 0, 0.5 and 1 respectively.

| "Thinking about the impact of parkrun on your health and wellbeing, to what extent has running or walking at parkrun changed:" | N | Mean | Standard deviation | Median | Minimum | Maximum |
|---|---|---|---|---|---|---|
| Your sense of personal achievement | 458 | 0.455 | 0.348 | 0.452 | -0.5 | 1 |
| Your fitness | 459 | 0.441 | 0.332 | 0.431 | -0.5 | 1 |
| Your physical health | 459 | 0.419 | 0.312 | 0.409 | -0.5 | 1 |
| The amount of time you spend outdoors | 451 | 0.317 | 0.319 | 0.299 | 0 | 1 |
| Your happiness | 458 | 0.314 | 0.305 | 0.302 | -0.5 | 1 |
| Your ability to be active in a safe environment | 462 | 0.314 | 0.324 | 0.295 | 0 | 1 |
| Your enjoyment of competing | 449 | 0.294 | 0.331 | 0.277 | -1 | 1 |
| Your mental health | 458 | 0.291 | 0.340 | 0.268 | -0.5 | 1 |
| Your confidence | 457 | 0.288 | 0.320 | 0.275 | -0.5 | 1 |
| How much you feel part of a community | 466 | 0.247 | 0.314 | 0.230 | -0.5 | 1 |
| Your ability to control your weight | 457 | 0.244 | 0.327 | 0.220 | -0.5 | 1 |
| Your overall lifestyle choices (e.g. diet & smoking) | 458 | 0.227 | 0.307 | 0.210 | -0.5 | 1 |
| The number of new people you meet | 458 | 0.198 | 0.302 | 0.178 | 0.0 | 1 |
| The amount of time you spend with friends | 455 | 0.149 | 0.265 | 0.139 | -0.5 | 1 |
| The amount of time you spend with family | 458 | 0.096 | 0.285 | 0.083 | -0.5 | 1 |
| To what extent has running or walking at parkrun changed your ability to manage your health condition, disability or illness | 458 | 0.117 | 0.271 | 0.104 | -0.5 | 1 |
| Mean impact | 468 | 0.271 | 0.201 | 0.254 | -0.312 | 1 |

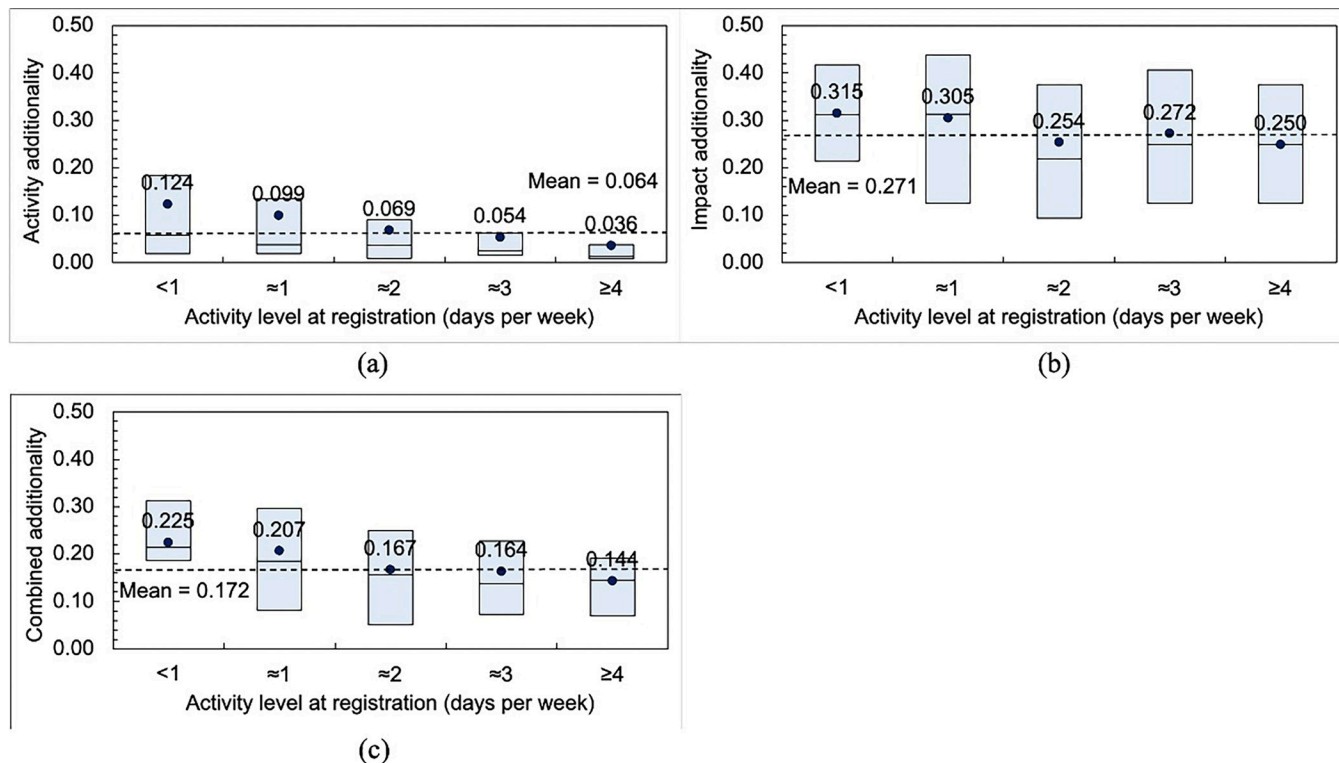

**Fig 4.** Weighted variables used to estimate the additionality of parkrun after six months of participation as a runner or walker: (a) parkrun activity additionality; (b) parkrun impact additionality; and (c) combined additionality (average of parkrun activity and impact additionality). Circles indicate means while boxes indicate interquartile ranges and medians.

## Calculation of WELLBYs

Table 4 shows mean WELLBYs for each method described by Eqs 1 to 5 and the total number of WELLBYs and its value for the full parkrun population of new registrants in 2019 (N = 400,167)

**Table 4. Mean WELLBYs segmented by activity level at registration using four methods to represent additionality of participation in parkrun.**

| | | Mean WELLBYs | | |
|---|---|---|---|---|
| Activity level at registration | Life satisfaction change | Method 1 Life satisfaction change and activity additionality | Method 2 Life satisfaction change and impact additionality | Method 3 Life satisfaction change and combined additionality |
| <1 | 0.769 | 0.029 | 0.232 | 0.159 |
| ≈1 | 0.423 | 0.035 | 0.129 | 0.082 |
| ≈2 | 0.297 | 0.011 | 0.082 | 0.042 |
| ≈3 | 0.056 | 0.002 | 0.039 | 0.020 |
| ≥4 | 0.174 | 0.021 | 0.074 | 0.048 |
| Mean | 0.257 | 0.016 | 0.087 | 0.053 |
| Total WELLBYs for 2019 population N = 400,167 | 51,341 | 3,101 | 17,407 | 10,544 |
| Value of 2019 population | £667.4m | £40.3m | £226.3m | £137.1m |
| Benefit-cost ratio | n/a | 16.7:1 | 98.5:1 | 59.3:1 |

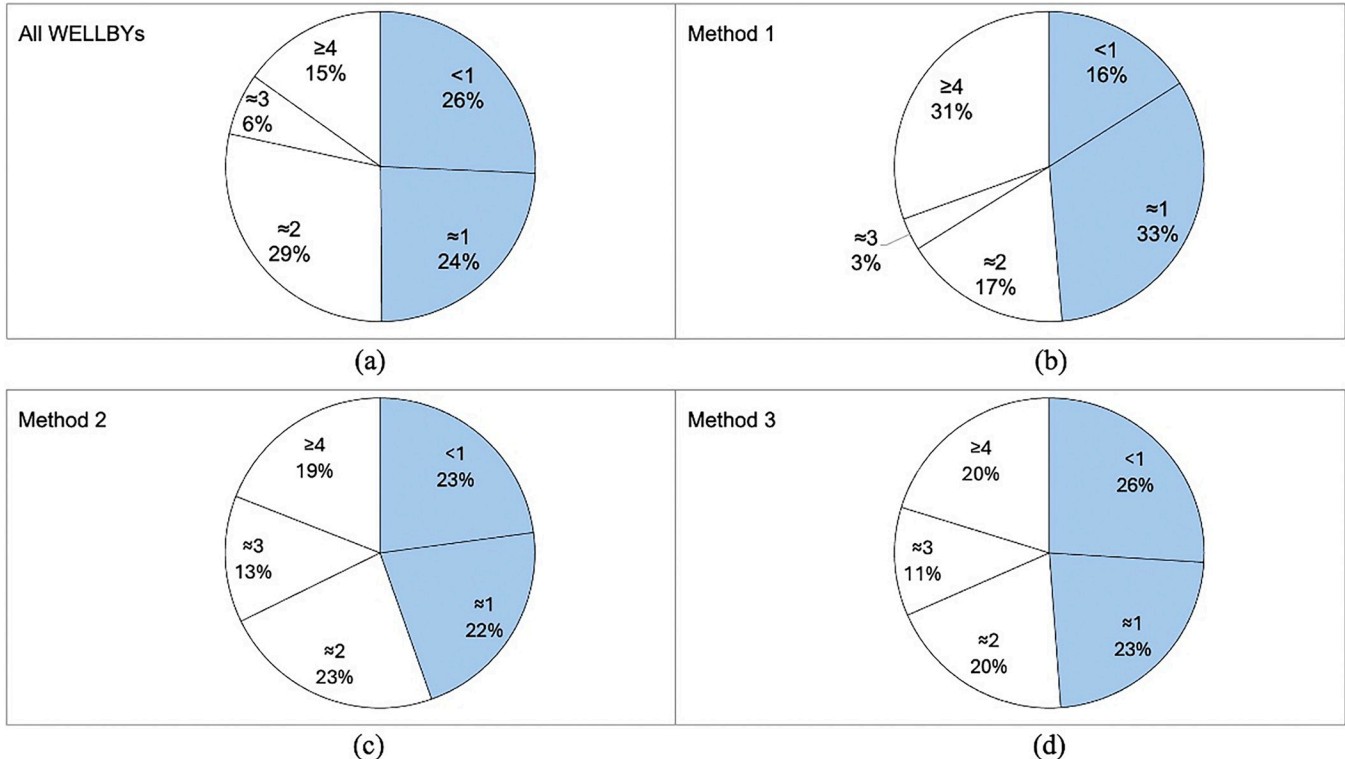

**Fig 5.** Proportion of WELLBYs for each activity level at registration for the four methods used to estimate parkrun additionality: (a) WELLBYS from all sources; (b) Method 1, WELLBYs using parkrun activity additionality; (c) Method 2, WELLBYs using parkrun impact additionality; and (d) Method 3, WELLBYs using a combined approach of parkrun activity and impact additionality. Blue segments indicate the two least active groups (<1 and ≈1).

The direct conversion of life satisfaction change to WELLBYs gave a mean value of 0.257 WELLBYs (per person per year), equating to a total of 51,341 WELLBYs for six months. At £13,000 per WELLBY this represented the total change, equivalent to £667.4m.

Using the three additionality methods, the benefit for six months change ranged from a minimum of 3,101 WELLBYs using activity additionality to 17,407 WELLBYs using impact additionality; the combined additionality benefit gave a value of 10,544 WELLBYs. This equated to values of £40.3 to £226.3m using activity and impact, and £137.1m using the combined approach.

Fig 5 shows the proportion of total WELLBYs in each activity level at registration; blue segments indicate the two least active groups (<1 and ≈1). These groups represent between 45 and 50% of the WELLBY value of parkrun, compared to Fig 1D which shows that they represent 23.3% of the parkrun population of new participants.

The minimum benefit-cost ratio of 16.7 to 1 was found using activity additionality; this increased to 98.5 to 1 using impact additionality with a ratio of 59.3 to 1 when activity and impact additionality were combined.

## Mediation model for parkrun

Fig 6 shows a possible mediation model between physical activity and life satisfaction following participation in parkrun for 6 months, as suggested by Maher et al. [36].

In the test of physical health as a mediator, activity was moderately correlated with physical health ($r = 0.227$, $p < 0.001$), and physical health was also strongly correlated with life

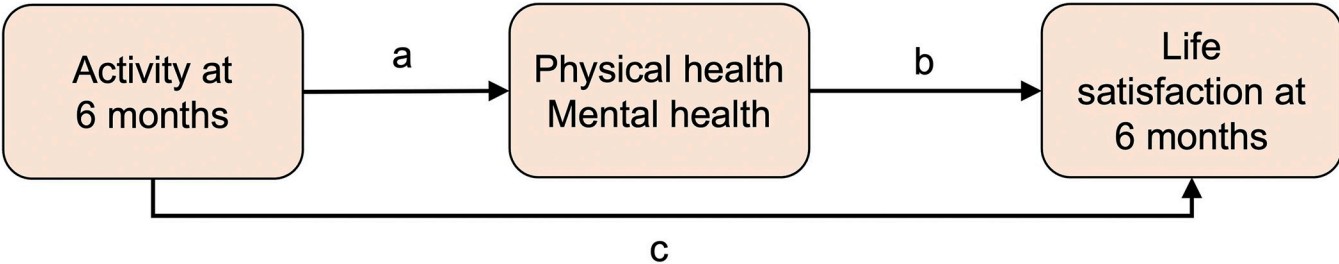

**Fig 6. Mediation model between physical activity and life satisfaction adapted from Maher et al. [36].**

satisfaction ($r = 0.541$, $p<0.001$). There was no correlation between activity and life satisfaction when physical health was controlled for, offering evidence of mediation.

In the test of mental health as a mediator, activity was not correlated with mental health, although mental health was strongly correlated with life satisfaction ($r = 0.609$, $p<0.001$). A weak correlation remained between activity and life satisfaction when mental health was controlled for ($r = 0.118$, $p = 0.013$). This did not show evidence of mediation.

Finally, in the test of both mental and physical health as combined mediators there was no correlation between activity and life satisfaction when they were both controlled for ($r = 0.024$, $p = 0.611$. This showed evidence of mediation when mental and physical health were combined.

## Discussion

### Change in life satisfaction due to participation in parkrun

The first research question posed in this study was whether participation in parkrun as a runner or walker impacted life satisfaction. The weighted survey results showed that after a median of two parkruns over a period of six months, newly registered parkrunners increased seasonally adjusted life satisfaction by a mean of 0.26. Life satisfaction went from below the UK average to above it with a change equivalent to about two-thirds of the drop experienced by the UK population during the COVID-19 pandemic, indicating that the change after parkrun participation is relatively large [33]. The least active were found to have the greatest increase in life satisfaction and may reflect previous findings showing that those with the least life satisfaction had the greatest capacity for change [21]. Despite this, the life satisfaction of the least active group was still the lowest and remained lower than the UK average, a feature found in other research in the UK [24].

A key question for a credible cost-effectiveness calculation is how much of the life satisfaction change can be attributed to parkrun. Considering only activity additionality to represent parkrun's impact, the attribution (converted to percentage) is estimated to be around 6.3%. At 12.4% for the least active, this is more than three times the 3.6% for the most active. A possible hypothesis for this is that parkrun may be new physical activity for the least active but might mostly replace other activities for the most active.

When considering the perceived impact of parkrun, this study suggests that parkrun might account for 27.1% of any life satisfaction change, with a gradual decrease from the least active at 31.5% to the most active at 25.0%, a difference of only 1.3 times. The smaller differences compared to activity additionality may be explained by a previous cross-sectional analysis of parkrun that found proportions reporting improvements to physical health, fitness and a sense of personal achievement, for instance, were quite similar regardless of demographic or activity level [8]. Taking a combination of both additionality approaches suggests an overall additionality of 17.2%, ranging from 22.5% for the least active to 14.4% to the most active.

## The cost-effectiveness of parkrun

The second research question addressed the cost-effectiveness of parkrun. Comparing to the cost of running parkrun for a 2019, this gave benefit-cost analysis ratios of 16.7 to 1, 98.5 to 1 and 59.3 to 1 using the activity, impact and combined methods respectively. This is between 2.8 and 16.7 times the 5.9 to 1 found for population-level interventions found previously [26]. Using a different method to calculate cost-effectiveness, a non-reviewed commission by parkrun suggested benefit-cost ratios of 120 to 1, much higher than those here [37]. The authors included benefits from volunteering, included children below 16 and arbitrarily assumed additionality of 25%, all leading to greater assumed benefits than here.

For transparency, it is prudent to consider the minimum possible cost-effectiveness using the data described in this study. Using the smallest benefit-cost ratio from activity additionality of 16.7 to 1 and the lower WELLBY confidence interval of £10,000 rather than £13,000, then a revised lowest benefit-cost ratio would be 12.8 to 1. This is still 2.2 times the value for population-level interventions [26].

The conclusion is that parkrun is at least twice as cost effective as other population-level physical activity interventions and potentially 16.7 times as high. In absolute terms, this makes parkrun a suitable population intervention with potential economic benefits for the 2019 parkrun population in the UK of up to £266.3m for a half year of participation. Previous research found that wellbeing benefits such as increased happiness and reduced stress could last at least 12 months [11] so it is possible that the potential economic benefits for 2019 would be double the values described here (although the benefit-cost ratios would stay the same). Future research should investigate whether any increase in life satisfaction is sustained or whether it follows the 'set-point' hypothesis and reduces over time. If parkrun only represents a proportion of the full increase in life satisfaction of parkrun participants, this prompts the question of which other factors influence life satisfaction: using the combined additionality approach, only 21.7% is attributed to parkrun so that 78.3% is still unaccounted for.

## Mediation effects by physical and mental health and implications for practice

Previous research has shown that the impact of physical activity on life satisfaction is likely to be mediated through physical and mental health [17, 22, 36]. The mediation analysis here also showed that physical health was a strong mediator between levels of activity and life satisfaction. Mental health alone was not found to be a mediator, although it had a notable effect when combined with physical health. The link between parkrun participation and life satisfaction, then likely stems from improvements to physical and possibly mental health.

Behaviour change to support this mediation at the population level requires careful design and one of the features of parkrun that supports this might be its 'framing'. Manuals for the delivery of effective psychotherapy emphasise the need for framing of any intervention in terms of its role, time, place and cost [38]. Parkrun addresses these as follows: its role is as a "*community event where you can walk, jog, run, volunteer or spectate*" [7]; it is at the same time every week (9am on a Saturday morning; 9.30am in Scotland and Northern Ireland); it is in the same place (a local park); and it is free.

Parkrun has also embraced methods of behaviour change identified by Abrahams and Michie in their taxonomy of behaviour change techniques [39, 40]. These include self-monitoring behaviour (getting a 5 km time), providing instruction (online and at the start of every event), encouragement (by marshals, onlookers and by email) and rewards (milestone t-shirts for number of parkruns completed). Finally, the repeated nature and longevity of parkrun might support participants to elevate physical and mental health over long periods of time

through both strong and weak social connections [12], keeping life satisfaction elevated and alleviating the issue of short-term change common for one-off mass events.

When total WELLBYs were calculated, almost half of the economic value came from the two least active categories, or participants doing about one or less than one bout of activity per week at registration. These are often from more deprived neighbourhoods and are traditionally those targeted most by government interventions [41]. This study shows that trying to attract those who are inactive is a logical approach since they experience the greatest increases in life satisfaction and the greatest additionality due to parkrun. It should also be noted, though, that those who are more active represent at least half the potential WELLBY benefit. Despite having lower increases in life satisfaction and additionality, they are more numerous. This appears to endorse the approach suggested in the Marmot review *Fair Society*, *Healthy Lives*, in which the principle of proportionate universalism was proposed: "*Actions should be universal, but with an intensity and a scale that is proportional to the level of disadvantage*" [42].

### Methodological considerations

Life satisfaction and other variables used in this study will contain subjective biases. While participants may reflect the parkrun population when weighted, the biases will still be present, and answers may not reflect those of the population as a whole. In particular, the sample was older and did more parkruns than the 2019 parkrun population; subjective assessments of impact will reflect this. The additionality of parkrun has been hypothesised to be a combination of activity and health and wellbeing impact. These (and other) estimates could be checked by comparing the WELLBY approach to a detailed social return on investment analysis [43].

### Conclusions

After six months, weighted, seasonally adjusted life satisfaction increased from a mean of 7.489 to 7.746, a change of 0.257. This took participants from below the UK average to above it. Life satisfaction was lowest for the least active and remained below the UK average after 6 months. It was estimated that 17.2% of life satisfaction could be attributed to the additionality of parkrun and both life satisfaction change and additionality was greatest for the least active. A calculation of total WELLBYs showed that almost half came from those doing one or less bout of activity per week at registration, and the overall benefit-cost ratio of parkrun was between 16.7 to 1 and 98.5 to 1. This was 2.8 to 16.7 times higher than other population-level interventions, suggesting parkrun is cost-effective. Physical health was found to be a mediator between activity and life satisfaction with mental health a possible additional factor. Features of parkrun that might be useful to other interventions were described: these include its framing (role, time, place, cost) and ability to forge both strong and weak social ties.

### Supporting information

**S1 Data. ONS wellbeing data.**
(XLSX)

**S1 Text. Health and Wellbeing survey.**
(PDF)

### Acknowledgments

Many thanks to Chrissie Wellington and Mike Graney at parkrun for helping create the original questionnaire and enabling the distribution of the survey. Thanks also to all the participants who filled out the surveys.

## Author Contributions

**Conceptualization:** Steve Haake.

**Data curation:** Steve Haake, Helen Quirk, Alice Bullas.

**Formal analysis:** Steve Haake.

**Funding acquisition:** Steve Haake, Helen Quirk.

**Investigation:** Steve Haake, Helen Quirk.

**Methodology:** Steve Haake.

**Project administration:** Helen Quirk, Alice Bullas.

**Resources:** Alice Bullas.

**Supervision:** Steve Haake, Alice Bullas.

**Validation:** Helen Quirk, Alice Bullas.

**Visualization:** Steve Haake.

**Writing – original draft:** Steve Haake.

**Writing – review & editing:** Steve Haake, Helen Quirk, Alice Bullas.

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
