## [Decision Letter · Decision Letter 0]

16 Jul 2024

PGPH-D-24-00708

Does participation in parkrun improve wellbeing and is it cost effective? A six-month study of parkrunners in the UK

Dear Dr. Haake,

Thank you for submitting your manuscript to PLOS Global Public Health. After careful consideration, we feel that it has merit but does not fully meet PLOS Global Public Health’s publication criteria as it currently stands. Therefore, we invite you to submit a revised version of the manuscript that addresses the points raised during the review process.

We kindly recommend:

structure the abstract according to the Journal requirement,double check the cited references to the literature,and pay particular attention to reviewers’ comments.

We look forward to receiving your revised manuscript.

Kind regards,

Hanna Nalecz, Ph.D.

Academic Editor

Journal Requirements:

Additional Editor Comments (if provided):

Reviewers' comments:

Reviewer's Responses to Questions

**Comments to the Author**

1. Does this manuscript meet PLOS Global Public Health’s publication criteria? Is the manuscript technically sound, and do the data support the conclusions? The manuscript must describe methodologically and ethically rigorous research with conclusions that are appropriately drawn based on the data presented.

Reviewer #1: Yes

2. Has the statistical analysis been performed appropriately and rigorously?

Reviewer #1: Yes

3. Have the authors made all data underlying the findings in their manuscript fully available (please refer to the Data Availability Statement at the start of the manuscript PDF file)?

Reviewer #1: Yes

4. Is the manuscript presented in an intelligible fashion and written in standard English?

Reviewer #1: Yes

5. Review Comments to the Author

Reviewer #1: This is not the first time topic of parkrun has featured in the PLoS journal and I wondered why this author did not reference the previous published parkrun study?

Title: The title of this study set in this style appear somewhat awkward almost like a newspaper article for such an important and intensive health promotion study seeking answers to the value of an intervention programme. Of course “Parkrun” is an intervention geared to improve quality of life in general having a time line for its evaluation. A simple title to say all this is readily available in the field of longitudinal study design contextualized in evaluating the initiative. A title revision is clearly required and this review suggests “Outcomes of participating in parkrun initiative in the United Kingdom. “

The major programme or study questions emerging from the parkrun initiative would be; to what extent would participating in the parkrun programme impact life satisfaction of participants, and secondly, what would be the level of cost-effectiveness of participating in the initiative?

Abstract: There is need for clarity in the statement of line 21: “...longitudinal study of 548 newly registered parkrunners asked the UK Office of National Statistics question on life satisfaction…”. Is the statement referring to the study or the participants? I suggest strongly that to establish certain clarity in the study argument, that the abstract should be presented in a structured way consistent with PLoS Global Public Health format: First paragraph should present the problem the study seeks to address, along with statement of justification warranting the study and the aim of the study. Paragraph two would present the methodological approaches involved and the results presented in the third paragraph with conclusion in the fourth paragraph.

6. PLOS authors have the option to publish the peer review history of their article (what does this mean?). If published, this will include your full peer review and any attached files.

**Do you want your identity to be public for this peer review?** For information about this choice, including consent withdrawal, please see our Privacy Policy.

Reviewer #1: **Yes: **Nnodimele Atulomah

---

## [Decision Letter · Decision Letter 1]

13 Aug 2024

The impact of parkrun on life satisfaction and its cost-effectiveness: a six-month study of parkrunners in the United Kingdom

PGPH-D-24-00708R1

Dear Professor Haake,

We are pleased to inform you that your manuscript 'The impact of parkrun on life satisfaction and its cost-effectiveness: a six-month study of parkrunners in the United Kingdom' has been provisionally accepted for publication in PLOS Global Public Health.

Best regards,

Hanna Nalecz, Ph.D.

Academic Editor

Reviewer Comments (if any, and for reference):

Reviewer's Responses to Questions

**Comments to the Author**

1. If the authors have adequately addressed your comments raised in a previous round of review and you feel that this manuscript is now acceptable for publication, you may indicate that here to bypass the “Comments to the Author” section, enter your conflict of interest statement in the “Confidential to Editor” section, and submit your "Accept" recommendation.

Reviewer #1: All comments have been addressed

2. Does this manuscript meet PLOS Global Public Health’s publication criteria? Is the manuscript technically sound, and do the data support the conclusions? The manuscript must describe methodologically and ethically rigorous research with conclusions that are appropriately drawn based on the data presented.

Reviewer #1: (No Response)

3. Has the statistical analysis been performed appropriately and rigorously?

Reviewer #1: (No Response)

4. Have the authors made all data underlying the findings in their manuscript fully available (please refer to the Data Availability Statement at the start of the manuscript PDF file)?

Reviewer #1: (No Response)

5. Is the manuscript presented in an intelligible fashion and written in standard English?

Reviewer #1: (No Response)

6. Review Comments to the Author

Reviewer #1: (No Response)

7. PLOS authors have the option to publish the peer review history of their article (what does this mean?). If published, this will include your full peer review and any attached files.

**Do you want your identity to be public for this peer review?** For information about this choice, including consent withdrawal, please see our Privacy Policy.

Reviewer #1: **Yes: **Nnodimele Atulomah
